# A snapshot on a journey from frustration to readiness–A qualitative pre-implementation exploration of readiness for technology adoption in Public Health Protection in Ireland

**Dorothea Ogmore Tilley** [1,2]*, **Brian McKeon**[1,2,3¤], **Nuha Ibrahim**[1], **Stephen H-F Macdonald**[1], **Marie Casey**[1,3]

**1** School of Medicine, Faculty of Education & Health Sciences, University of Limerick, Co. Limerick, Ireland, **2** Health Research Institute, University of Limerick, Co. Limerick, Ireland, **3** Department of Public Health Mid-West, Health Service Executive, Co. Limerick, Ireland

¤ Current address: National Ambulance Service, Health Service Executive, Co. Limerick, Ireland.
* thea.tilley@ul.ie

**Data Availability Statement:** The authors did not specifically ask participants for consent to have their full anonymised transcripts made publicly available due to the risk of identification, so the

## Abstract

In an era of emergent infectious disease, the timely and efficient management of disease outbreaks is critical to public health protection. Integrated technologies for case and incident management (CIM) collect real-time health intelligence for decision making in Public Health. In Ireland, a Public Health reform program is preparing for implementation of a health information system for health protection. Project implementers seek to document and understand the readiness and willingness of future users to adopt the new system, prior to system procurement and implementation. Qualitative key informant interviews were conducted (n = 8) with Public Health personnel from a single regional department of Public Health representing medical, nursing, disease surveillance and administrative roles, at managerial and staff levels. A qualitative thematic analysis was performed. Participants were frustrated by weaknesses in the current practice of CIM and were ready and willing to adopt a digital CIM system if it met their needs. However, they were frustrated by lack of clear timelines. We identified 7 enablers and 3 barriers to readiness and willingness to adopt a CIM system. 'Newness of the workforce' was the main enabler of readiness and willingness, while 'lack of knowledge and familiarity with system' was the main barrier to readiness and willingness. Experiences during the COVID-19 pandemic gave a clear understanding of the problems and need for a digital CIM system and the reform program facilitated a culture of change, readying the workforce for the new health information system. New members of the Public Health departments are a likely ready and eager cohort for adoption of a modern, 'fit for purpose' CIM system and the execution of implementation will likely determine how ready and willing the wider network of departments will be to adopt a national CIMS.

data policy exception related to privacy concerns pertains in this case. Excerpts of data from this study are available on request to University of Limerick Hospital Group Research Ethics Committee (reference 055/2022). Email: ULHGResearchEthicsandClinicalTrials@hse.ie. Address: Quality & Safety Department, UL Hospitals Group, HSE Unit 2, Loughmore Avenue, Raheen Business Park, Limerick. V94 P7X9.

**Funding:** The author(s) received no specific funding for this work.

**Competing interests:** The authors have declared that no competing interests exist.

## Author summary

In an era of emergent infectious disease, timely and efficient outbreak management is critical to public health protection. The COVID-19 pandemic highlighted this need at a global scale. Integrated health technologies for disease monitoring, laboratory results, contact tracing, and case management facilitate real-time data collection for decision making. The Irish health service is preparing to implement a digital national health protection system. To ensure successful technology adoption, the organisation must be ready for change. We interviewed a multidisciplinary group—public health doctors, nurses, surveillance scientists and administrators—working in health protection and disease surveillance. All study participants had a clear understanding of the problems with current practices and were eager to adopt a new system to help them work more efficiently and provide a better service to the public. However, they were concerned about the capabilities of a new system and believed a lack of familiarity with the new system made staff less ready and willing to adopt the technology. Nonetheless, as Public Health was a field that had witnessed great change during the COVID-19 pandemic and employed many new people, they felt the 'newness' of the workforce could enhance readiness for change as an organisation.

## Introduction

Disease surveillance, reporting and case management are essential to protecting health from biological and environmental hazards [1]. In 1998, African nations recognised the importance of integrated disease surveillance and response systems before more resourced and less infectious disease burdened neighbours [2,3]. The design and implementation of these systems in low- and middle-income countries now informs health protection case and incident management system (CIMS) use worldwide [4–6]. The optimal surveillance and response systems are multifunctional with integration of case management, disease surveillance, contact tracing and laboratory results [5,7,8]. Furthermore, bi-directional communication channels allow coordinated use of resources, critical for effective resource planning during times of high pressure e.g. during a disease outbreak [7].

In Ireland, reporting and investigation of suspect incidents of biological and environmental exposure is the responsibility of the national health service (Health Service Executive—HSE), its regional Departments of Public Health (DoPHs) and their health protection medical, nursing and surveillance personnel. The current case and incident management (CIM) practice is not regionally standardised nor electronically integrated into laboratory reporting systems or the national disease surveillance centre. Critically, regional DoPH practices in CIM are ad hoc, using a combination of paper and electronic records, risking inefficient use of resources and weaknesses in data management and medico-legal compliance. While a national health information system for infectious disease surveillance was established in 2004, the Computerised Infectious Disease Reporting (CIDR) system, this system is limited to notifiable diseases and excludes environmental and chemical/radiological incidents [9,10]. Additionally, it does not contain the clinical notes supporting case and incident management. More recently, a contact tracing system was developed using a customer relationship management platform, but use of this system continues to be limited to COVID-19 alone.

Thus, and in light of the COVID-19 pandemic, there has been renewed focus on the introduction of a fully integrated health protection CIMS that is adapted to the Irish context [11]. However, introducing new technologies into healthcare contexts is challenging [12]. It is helped by participatory approaches and applying psychosocial theories of behavioural change

that have evolved to theories of technology adoption [13,14]. The Technology Acceptance Model (TAM) [15], Unified Theory of Acceptance and Technology Use (UTAUT) [16], and Diffusion of Innovation (DOI) [17] are 3 of the commonly used theories and models for managing and predicting adoption of new health information systems (HIS) [14,18]. In TAM and UTAUT, 'behavioural intent to use'—an individual's motivation or willingness to exert effort to perform the target behaviour (to use the system)—is the main determinant of 'actual use' [19]. Both models use a version of 'perceived ease of use' and 'perceived usefulness' as antecedents to 'behavioural intent to use', with TAM also including 'attitude' and UTAUT using 'social influence, facilitating conditions, age, gender, experience and voluntariness of use' [19]. In contrast, the persuasion phase of DOI links users' perspectives on: the technology's relative advantage over previous practice; compatibility with users' values, past experiences and needs; the complexity and trialability of the technology; and observability of the results of the innovation to the rate of adoption [17]. A network of 'activated' peers can then collectively influence others in the social context of the innovation to adopt ('the diffusion effect') [17]. On top of such models, in the context of applying new technologies to public sector and or health related settings, system level variables should also be added that go beyond user perceptions—e.g. teamwork, or user behaviours such as 'workarounds' and learning [13]. New approaches are recommended and include understanding and reconciling multiple user needs, temporally evaluating HIS implementation and applying theories that reflect more complexity of organisation and user [13].

At an organisational level, the practice of Public Health in Ireland is currently undergoing reform with the transition from medical only to multidisciplinary team approaches [20]. Furthermore, DoPH staff have experienced great change, sometimes accompanied by innovation, during the COVID-19 pandemic. With this backdrop, this study is the first step in a journey to document and evaluate implementation of a national digital CIMS for health protection.

## Aim

This study aimed to document and understand future users' attitudes and behavioural intent, in the form of readiness and willingness, to adopt a new CIMS into Public Health practice, prior to system procurement and implementation.

## Objectives

- Using qualitative key informant interviews from a multidisciplinary group of future users, the study builds a rich picture of future user readiness and willingness to adopt the technology.

- Informed by theories and models of technology adoption, it explored potential factors that may influence future user's adoption of the technology.

- Specifically, it explored future users' past experiences and perceptions of change; their experiences and perceptions of the current practice of CIM, including digital literacy competencies; their perceptions and experiences to date in the plan to change the practice of CIM; and their perceptions of readiness and willingness to adopt the new technology.

- It identifies barriers and enablers which can be leveraged to increase readiness and willingness across the organisation during implementation.

## Methods

### Ethics statement

Ethical approval was obtained from the University of Limerick Hospital Group Research Ethics Committee (REC number 055/2022) and written informed consent to participate was provided by all participants. All authors consent to have this research published and the participants have consented to the dissemination of findings.

### Study design

This study adopted an observational, qualitative descriptive design. It used a qualitative descriptive approach with elements of phenomenological and narrative approaches. This facilitated exploration and description of participants' attitudes and behavioural intent to adopt a new HIS for health protection based on their past experiences of change and their present views on, and experiences to date in the planned reform of CIM in Public Health. It was guided, but not constrained by, theories of technology adoption [21] and organisational readiness for change [22] so as to ensure exploration of dimensions known to influence the adoption of new technologies. The findings arise from a constructivist research paradigm [23]. The study was conducted and was reported according to the EQUATOR network recommended Standards for Reporting of Qualitative Research (S1 Text) [24].

### Researcher characteristics & reflexivity

DOT performed the interviews, analysis and interpretation of results, and was not known to participants before the interviews. There is a professional relationship between the project lead (MC) who is a Consultant Specialist in Public Health, the co-investigator BMK (General Manager) and the participants as they are all employees in the same organisation (DoPH Mid-West, HSE) and are known to each other. MC and BMK are also members of the National Program Team for Outbreak CIMS, with responsibility for selection and implementation of a CIMS–MC: clinical lead and BMK: business manager. To counter potential researcher bias, an inductive coding approach was used to capture participants perspectives. Interviews were conducted online and presumed to take place at the participants' workplace and thus their contributions may have been influenced by these relationships and the setting. Having an independent researcher (DOT) conduct the remote interviews and analysis is hoped to alleviate these potential influences on the findings perspectives.

### Study context

This study contributes to the planned implementation of a standardised national CIMS across all Irish DoPHs and is embedded in a strategic plan to reform Public Health practice in Ireland [20]. A timeline of the contextual events surrounding this study, from 2015 to 2023, are outlined in Fig 1. Events include declaration of the COVID-19 pandemic, a cyber-attack on the health service IT systems [25], transition to multidisciplinary team working, and demonstrations of candidate systems. This study may inform the design of a future national survey to assess the readiness and willingness of members of all regional DoPHs to adopt a national CIMS.

### Study area & period

This present study took place over the period of June 2022-May 2023. Interviews were conducted remotely from DOT's home (Ireland) and presumed to take place at the participants' workplace (the Department of Public Health Mid-West, Limerick, Ireland) but it is possible some interviews took place at other locations due to remote working policies at the time of investigation.

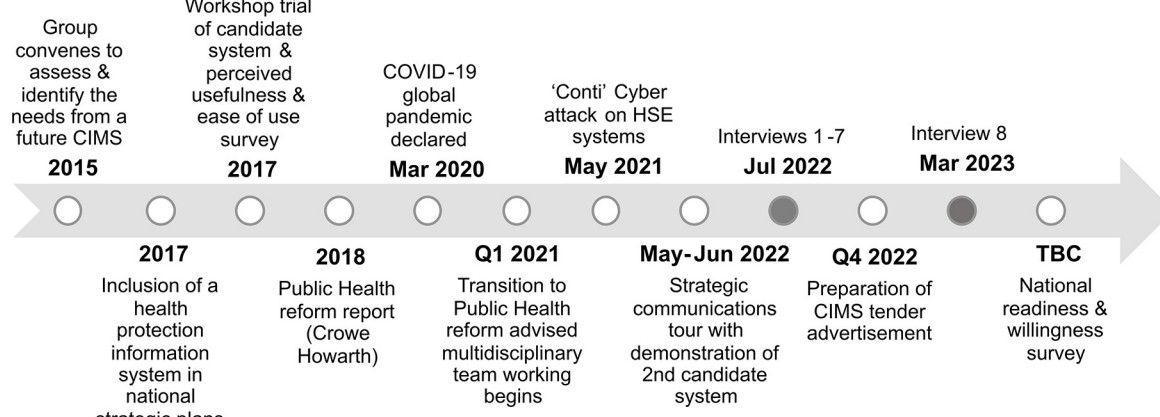

**Fig 1. Study positioning in the timeline of contextual events.** Schematic representation of the timeline from the first work on a national CIMS for health protection to key informant research interviews & a future national readiness & willingness survey. CIMS–case & incident management system; HSE–Health Service Executive; Q1 –first quarter (of year); Q4 –fourth quarter (of year); TBC–to be confirmed.

## Study population

The population in this study were employees of the Irish national health service, and were members of a regional department, the Department of Public Health Mid-West (Limerick, Ireland). The potential pool of participants in this single site was 45, spread across 4 categories: medical, nursing, surveillance and administration. No social or demographic information was available to researchers to describe the source population.

## Sampling strategy

The sampling objective was to obtain the perspectives of a rich and diverse set of future users. These user groups were identified through a) discussions with the researchers MC and BMK who had existing knowledge of which roles were involved in the practice of CIM and disease surveillance or interacted with its systems and b) consultation of organisational charts which described the roles and levels in the organisation. Early participants also suggested future user groups which confirmed the already identified groups. Due to the limited number of people in the department and availability due to the workload during a pandemic, achieving data saturation was not part of the sampling strategy. Thus, a minimum sample size of 8 was determined, with researchers aiming to recruit 2 persons representing each of the 4 categories of DoPH member, in senior (managerial) and junior (staff grade) roles respectively. Purposive sampling was used to achieve this as it allowed selection of participants based on role and seniority level. Candidates were identified by BMK who then shared contact details with the interviewer, DOT. DOT invited candidates to participate by email including a consent form and participant information sheet outlining the purpose, risks, impact of the study, data use and their role in this phase of the wider project implementation plan.

## Study participants

A total of 10 participants were invited and 2 declined to participate due to other commitments (n = 8). During the period June-July 2022, 7 participants were recruited representing medical, nursing and administrative roles. The final participant, representing the surveillance group perspective, was recruited May 2023. Medical, nursing and administrative perspectives were

**Table 1. Distribution of participants across organisational group & seniority level.**

| Organisational group | Number of participants |
|---|---|
| Medical | 2 |
| Nursing | 2 |
| Surveillance | 1 |
| Administration | 3 |
| **Seniority level** | |
| Senior (managerial) | 3 |
| Junior (staff grade) | 5 |
| Total participants | 8 |

Total participants (n = 8) categorised by organisational role or seniority level. To maintain participants confidentiality, 'Seniority level' is an independent descriptor not linked to organisational group to avoid identifying individuals' responses as there are limited numbers in the department in that position and role.

obtained for managerial and staff grade roles (Table 1). Perspectives from a single seniority level for the surveillance key informant was obtained. To further improve confidentiality in a small department, demographic characteristics were not collected. However, in Ireland, females are over represented in Public Health medicine, nursing and healthcare administrative roles and the sample group reflects this [20, 26].

## Ethical issues pertaining to human subjects

Ethical approval to conduct this study was obtained from the University of Limerick Hospital Group Research Ethics Committee (REC reference number 055/2022. Participation of employees in this research was voluntary and prior to interview they provided written informed consent to participate and have the anonymised findings of the study disseminated. However, as the study was conducted in a workplace setting and led by a senior co-worker, this raised additional ethical considerations [27] and steps were taken to ensure participant confidentiality and welfare. All communications with participants were through an independent researcher (DOT), outside of the organisation. The voluntary nature of participation, participation/non-participation without the knowledge of MC and BMK, options to withdraw and the methods of de-identification of data to ensure confidentiality were communicated in the participant information sheet during the informed consent process and verbally prior to commencing the interview. DOT was the sole researcher with access to the transcripts in raw or anonymised forms. Only anonymised aggregated data were shared with MC or BMK to maintain participant confidentiality. In reporting the anonymised findings, participants' data, quotations, have not been assigned to specific roles but to participants' assigned random letter or to organisational group (administrative/nursing/medical/surveillance), or seniority level (managerial/staff) to convey their perspective when appropriate. When highlighting seniority or organisational group perspectives participant letters were not used to prevent identification of participants by MC and BMK and colleagues.

## Data collection

Data for this study was collected through key informant (KI) interviews conducted online, using virtual meeting software. A semi-structured interview guide was developed based on topics in technology adoption theory, organisational readiness, and contextual factors, to explore individual and perceived collective perceptions of the plan to introduce a national CIMS, the perceived readiness and willingness of the DoPH to adopt a CIMS, and how best to implement it (a subsequent publication will examine participant preferences for

implementation and approaches to enhance readiness and willingness). The interview guide explored the following: 1) time in the department, digital literacy; 2) understanding and assessment of the problems that the technological solution needs to address; 3) perception of usefulness of the technological solution; 4) experience of the planned change to date—level of awareness, channels of communication, personal and collective response to specific change, reasons for response; 5) suggestions for implementation informed by past experiences of change–personal experience of change, facilitating conditions; 6) perceptions of readiness and willingness and barriers and enablers to readiness and willingness; 7) perceptions of cohesion and involvement in change; and 8) the effect of COVID-19 on perceptions of change. DOT tested the guide informally with peers and changes were made to make questions open and improve the flow of the interview (S2 Text).

KI interviews were conducted by DOT, virtually, over two periods: a two-week period (27-06-2022 to 08-07-2022) for interviews 1–7 and a single day in March 2023 for the final KI interview (interview 8). Challenges recruiting the final KI and availability of the interviewing researcher prevented collection of the data over a single discrete period. Interviews were recorded and automatically transcribed using Zoom (n = 1) or Microsoft Teams (n = 7). The interview guide was piloted with one participant to see if questions and length of interview were appropriate (45 min). No changes were made to the interview guide as a result and so this interview was included in the subsequent processing and analysis. A single pilot was performed due to limited availability of participants. Seven further interviews were conducted and the average interview time was 65 ± 23 min (± standard deviation), range 38–112 min.

## Data processing

Automatic transcription functions in Zoom and Microsoft Teams were used to generate a text file of the interview that was imported into Microsoft Word and was checked for accuracy by reviewing it in parallel to the interview recording. The transcript was corrected accordingly and the original interview recording was deleted. All files containing individualised raw data were pseudo anonymised and password protected with a key held in a separate password protected file by DOT to allow revision of transcripts by participants and withdrawal of participation if requested. Data was (pseudo) anonymised by replacement of the participant's name with a randomly generated letter and removal of personal data and any identifying details relating to role title, previous roles or jobs, locations, people etc.

## Data analysis

To create a rich picture of the diverse set of DoPH members perspectives and gain insight into the readiness of participants to adopt a new HIS, we combined inductive and deductive approaches (using technology adoption theories). This is compatible with the methodology of reflexive thematic analysis [28, 29] and was performed as subsequently outlined. Data familiarisation was performed in parallel to transcript correction. An iterative coding process was adopted and coding was performed using Nvivo qualitative analysis software (version 1.6.1 [1137], QSR international, United States). Coding was guided by the question 'what knowledge, experience, feelings and suggestions had the participant on the introduction of a digital CIMS to network of DoPHs and what insights from their past experiences of change might improve the experience of implementation?'. Transcripts were coded by sentence or paragraph as appropriate to the length of relevant text. Initial codes were generated based on open-coding from interview 1 and reorganised into themes. Then interviews 2–5 were coded and reorganised into existing themes and codes and new ones added where required. Interviews 6 and 7 were processed similarly. Themes and subthemes were then reviewed—checked, recoded and

renamed as necessary. A preliminary thematic map was prepared. The final interview was conducted and coded to the themes and new codes and themes added where necessary before reorganisation a final time and a final thematic map was prepared.

### Techniques to enhance trustworthiness

All participants were given the option to review their interview transcripts and two took up the option to receive a copy and no changes were made. Member checking was performed by inviting participants to review a personalised document collating their quotations in the context of the resulting themes and asking them to correct or confirm the interpretation and use of their quotes in the text. They were also invited to highlight for further anonymisation, any information they thought might make them identifiable to colleagues (all personal data was removed prior to analysis). 5/8 confirmed the interpretation of quotes and did not indicate any corrections to preserve confidentiality. Three participants did not respond. To increase transparency, data tables of illustrative quotes for themes and subthemes are reported in the supplemental information (S1 Table–S7 Table). These data tables were also included in the member checking process.

## Findings

A thematic map of the main themes from the analysis of the interviews is presented in Fig 2

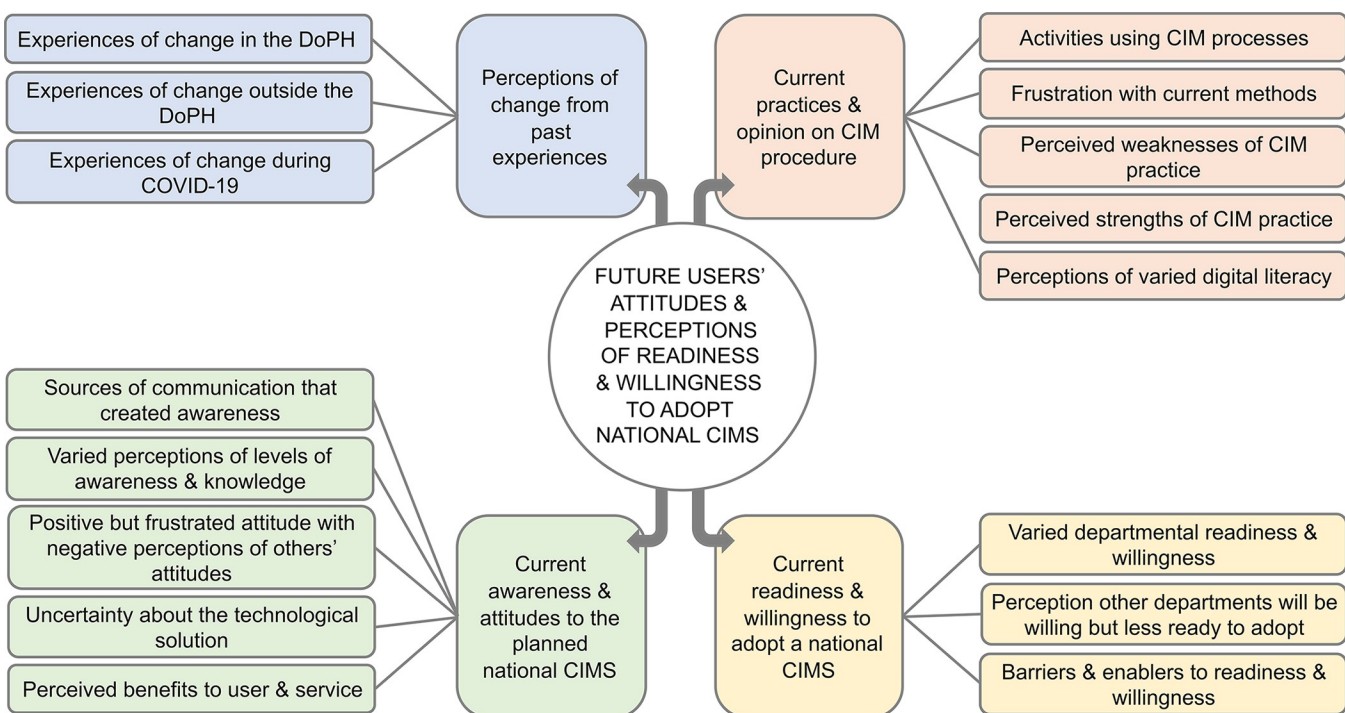

**Fig 2. Thematic map representing the overarching categories and main themes from thematic analysis.** The thematic map outlines the research focus with 4 overarching categories displayed in different colours and indicated emanating from the centre. These 4 categories are ordered as follows 1) perceptions of change from past experiences, 2) current practice and opinion on case and incident management (CIM) procedure, 3) current awareness and attitudes to the planned national CIM system (CIMS) and 4) current readiness and willingness to adopt a national CIMS. Main themes of each category are indicated and expanded on in other figures.

## Perceptions of change from past experiences

The majority of participants were relatively new to the department, joining the department after March 2020 during the response to the COVID-19 pandemic. Most individuals' experiences and deeper understanding of change came from outside the DoPH, mostly in health or public service-related settings. There was a mix of those who had worked in the health service for some years and for whom this was their first role the health service. In the category perceptions of change from past experiences, themes were: experiences of change in the DoPH (positive and negative); experiences of change outside the DoPH (positive and negative); and experiences of change during COVID-19—Fig 3.

**Experiences of change in the DoPH.**  In their current roles, most participants indicated positive experiences of change, with changes made in response to staff needs, good engagement and communication during recent changes, and, when change is needed, there are open communication channels.

> 'we have been involved, . . . I think the department, . . . it has been very engaging with all it's staff changes on pages, so we have to move office and I think . . . while our own was being done up there was very good . . . communication done with that. There was a representative from each . . . grade or position, peer group, peer group. That worked very well.' (Participant H)

> 'there is a good pathway . . . all the way up to [the Director] it's there . . . you can make your way up to [them] with information if you need to' (Participant E).

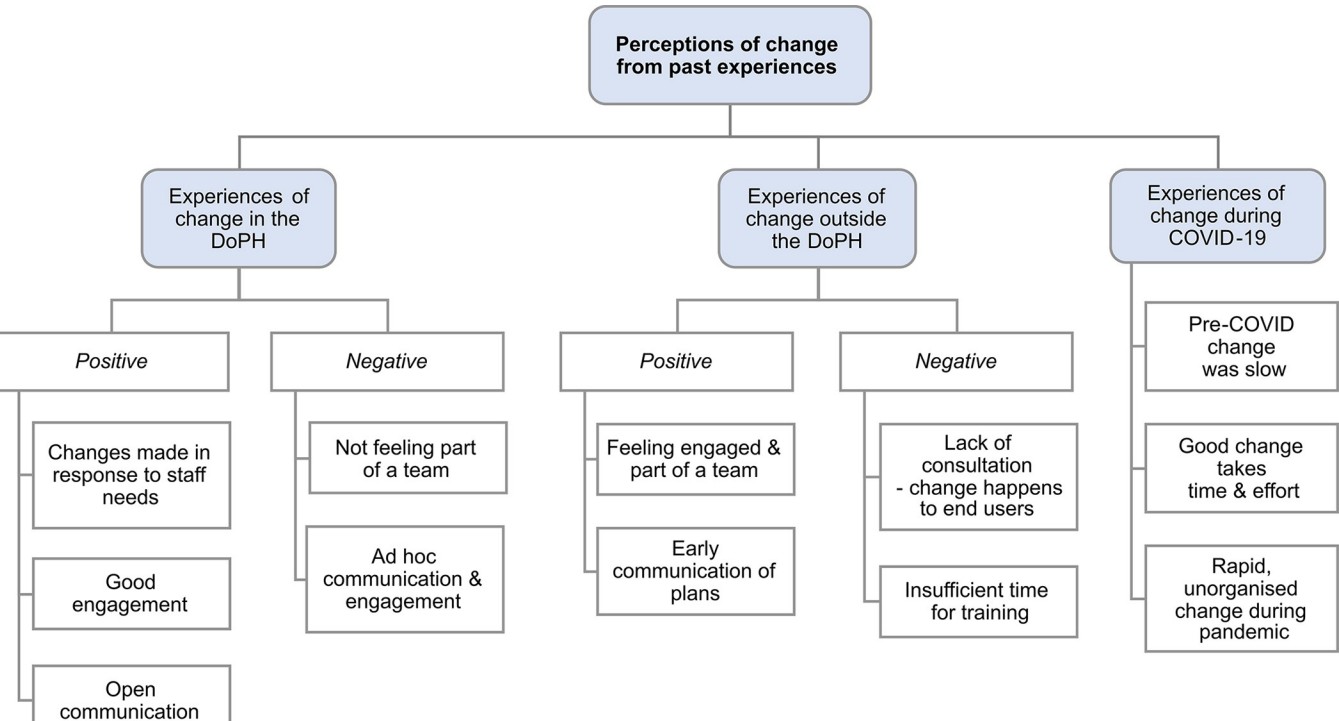

**Fig 3. Summary the themes and subthemes in perceptions of change from past experiences.** Themes in participants' experiences of change were experiences of change in the Department of Public Health (DoPH), experiences of change outside the DoPH (in previous roles) and experiences of change during COVID-19. Sub-themes are grouped in white boxes and an organising theme, positive or negative included where relevant. Illustrative quotes for all sub-themes are presented in the supplementary information (S1 Table).

However, some participants felt communication and involvement in change was ad hoc and not all participants felt part of a team and emphasised the importance of a team approach to implement the CIMS.

**Experiences of change outside the DoPH.**   Outside of the department, participants had a mix of positive and negative experiences of change. Historically, change was noted as quite 'hierarchical', with change happening to end users. There was a lack of consultation with end users, particularly around their input into the decided change, insufficient time for training and sometimes communication was poor, with information or changes not being communicated in a timely manner. In contrast, changes resulting from new policy and legal obligations with deadlines were communicated to affected users early and participants felt prepared despite having no choice as to whether the change would be implemented. Similarly, having a pre-implementation agreement plan documenting responsibilities and timelines for all stakeholders made change easier when implementing new national health information systems.

'All of that was decided . . . at the outset and . . . the responsibilities . . ., all of that was . . . part of the agreement plan, that it was all pre-agreed and sorted' (Participant C).

Feeling engaged and part of a team was important for positive experiences and some participants expressed personal enthusiasm for change. Notably, poor experience of change did not equate to a bad or unsuccessful change.

'I would say change experience hasn't been that positive, even if the change is positive' (Participant B).

**Experiences of change during COVID-19.**   Achieving change prior to COVID-19 was considered slow, and it was appreciated that time and effort is needed for change. In contrast, during the pandemic, rapid, unorganised, and abnormal change occurred but it demonstrated the capacity of the organisation and its staff for change.

'things that . . . were perceived could never change . . ."this is always the way things are done", overnight changed' (Participant A).

## Current practices & opinion on CIM procedure

In the category of current practices and opinion of CIM procedure, themes were: activities in CIM processes; frustration with current methods, perceived weaknesses of CIM practice; perceived strengths of CIM practice; and perceptions of varied digital literacy (Fig 4).

**Activities using CIM processes.**   All participants used or interacted with the existing systems/processes for CIM to carry out their roles and responsibilities. The range of activities these interactions covered was wide. Participants used the current CIM processes not only for the performance of clinical CIM and disease surveillance, but also for non-CIM and disease surveillance purposes, including administrative, supportive, quality assurance and reporting activities.

**Frustration with current methods.**   The overall experience of the current practice of CIM was one of frustration, due to the many perceived weaknesses in the systems and processes.

'it's incredibly frustrating. They're very old and they're not really fit for purpose . . . they're prone to error . . . they're very hard to navigate. It's not clear. So from the technology that I use in my everyday life . . . it's really behind. Behind the times" (Participant B).

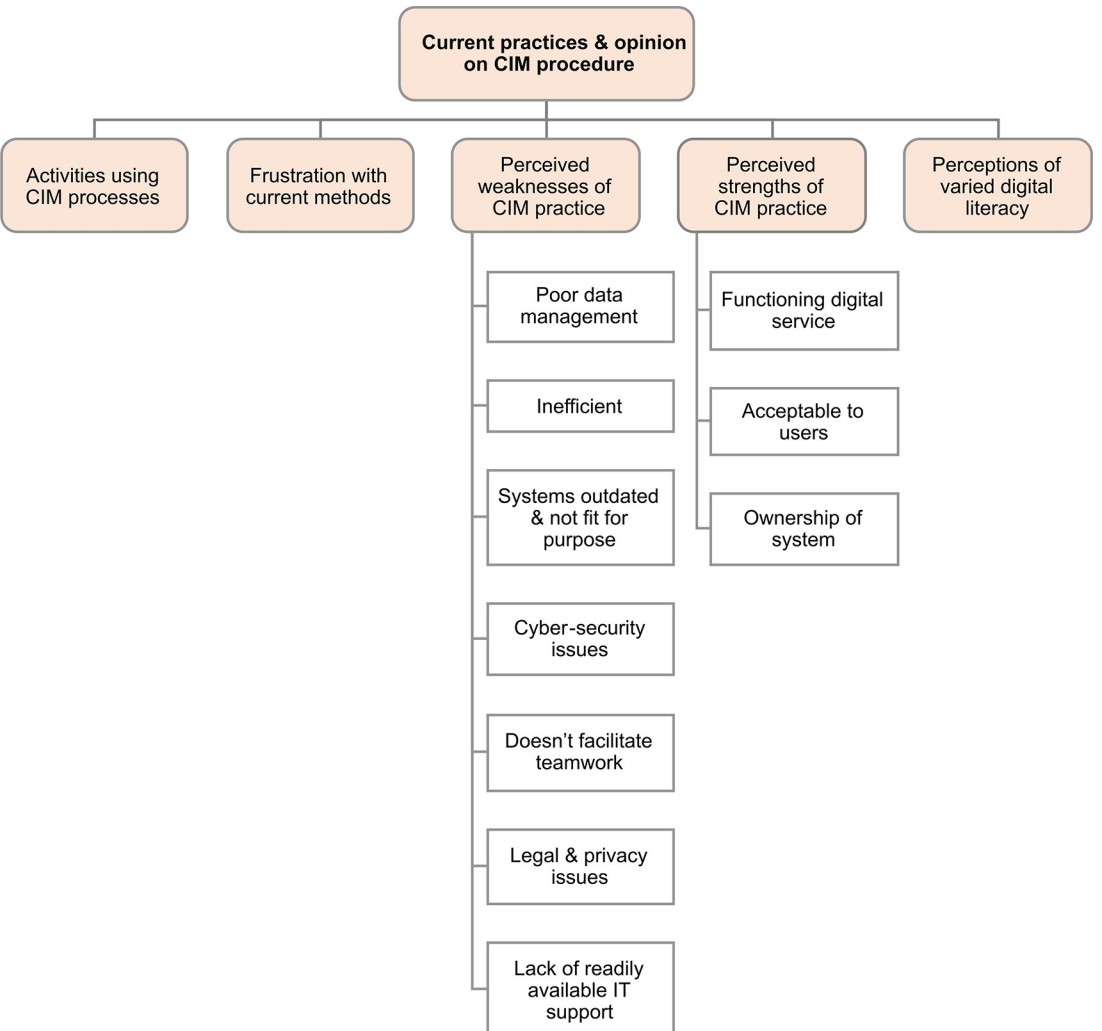

**Fig 4. Summary of themes and subthemes in current practices & opinion on CIM procedure.** Current practices and opinion on CIM procedure has 5 main themes. Further subthemes for perceived strengths and weaknesses are grouped in white boxes and Illustrative quotes for these and codes detailing specific weaknesses and strengths of the current practice are presented in the supplementary information (S2 Table & S3 Table). CIM—case and incident management; IT–information technology.

**Perceived weaknesses of CIM practice.** The systems were inefficient and considered not fit for modern Public Health practice and disease surveillance. There were multiple work-streams for receiving disease notifications, with disease specific and general systems, laboratory systems and postal notifications. Data was heavily reliant on human input and, as a result, was error prone and lacked automated alerts for inconsistencies or linking of cases. A cyber-attack in 2020 demonstrated security and data storage issues. The systems didn't facilitate teamwork, with some systems designed of one active user at a time. There was a lack of readily available IT support on site and users spent significant time logging issues with the national IT support service. In addition to CIM, administrative participants used the data collection for external reporting and legally mandated information requests and reported that this task required that users directly involved in CIM (clinical and surveillance roles), identify and collect the data for administrators which was both frustrating and time consuming. Critically, poor data management and difficulty finding data were the most commonly described weaknesses.

'we could open up an outbreak case, a situation created and then two days later, end up opening it up, creating a whole new one for the same situation because we couldn't find the original one because . . . a search function wasn't clever. Or . . . somebody misspelled something' (Participant E).

**Perceived strengths of CIM practice.**   Despite the many weaknesses, participants acknowledged some strengths of the current methods. Importantly, it was a functioning digital service and was familiar and acceptable to users. It was considered a simple and transparent system, training was easy and it supported remote working. There was also a sense of ownership for aspects that had been developed in house.

'at least, everything is filed electronically, and . . . we don't have large . . . paper systems and filing cabinets . . . paper trail is now an electronic trail . . . so that's a good thing and it leads to probably more transparency' (Participant H).

**Perceptions of varied digital literacy.**   Most participants acknowledged a mix of digital literacy skills and experience levels across the department and noted that more support would be welcome. Some self-evaluated as having lower digital literacy compared to colleagues but indicated comfort with the Microsoft suite suggesting higher skills with familiar technologies. Overall, the participants appeared to have average to high digital literacy when combined with being comfortable learning new technologies and there was a perception of 'an upward trend' in digital skill proficiency in the department. Administrative and surveillance participants indicated higher digital literacy, with no fear of learning new systems if 'proper training is provided' as they are 'used to' working with digital software packages. However, it was thought clinical areas may find it more challenging and relating challenges to digital literacy to age, experience and role. Participants from both administration and clinical groups expressed that 'more support' would help, with nursing acknowledging a culture of shared learning of systems in their discipline. In contrast, another participant considered that the nature of the department—its knowledge of outbreak management systems (OMSs) and its small size—facilitated in-house support for digital literacy that would be needed for the CIMS.

'some . . . sections . . . maybe nursing and medical, wouldn't have the same level of experience . . . and probably would need that bit more support' (Participant A).

'we're all different ages and different technologically abled, and I think . . . some people find it very easy and . . . some people find it a bit more challenging so support to hand would be good' (Participant H).

## Current awareness & attitudes to the planned national CIMS

In the category of current awareness and attitudes to the planned national CIMS, themes were: sources of communication that created awareness; varied perceptions of levels of awareness and knowledge; positive but frustrated attitude with negative perceptions of others' attitudes; uncertainty about the technological solution; and perceived benefits to user & service (Fig 5).

**Sources of communication that created awareness.**   Participants initial awareness of the plan to introduce a national digital CIMS came through 3 main routes: casual discussion with colleagues; management meetings; and the national Public Health reform plan and presentations related to it. There was contrasting evidence that the planned change was discussed in peer groups—"A little bit . . . kind of informally. Just [with] my colleagues. They'd be . . . of

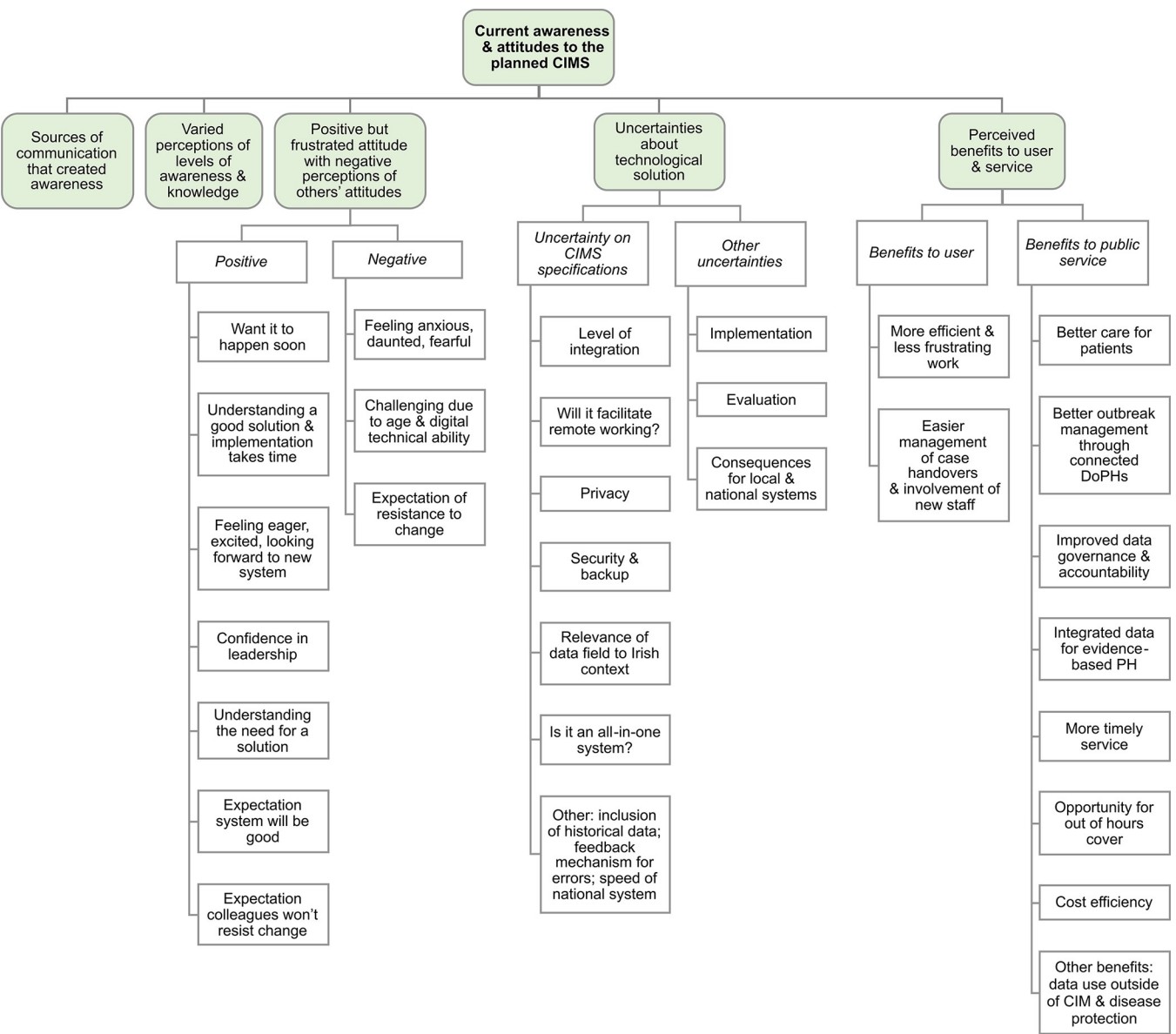

**Fig 5. Summary of the themes and subthemes in current awareness and attitudes to the planned CIMS.** Current awareness and attitudes to the planned CIMS has 5 main themes. Further subthemes for positive but frustrated attitude with negative perceptions of others attitudes (organised by positive and negative), uncertainty about the technological solution (organised by uncertainty on CIMS specifications and other uncertainties), and perceived benefits to user and service (organised by benefits to user, benefits to service and other benefits) are grouped below in white boxes and Illustrative quotes for these are presented in the supplementary information (S4 Table, S5 Table and S6 Table). DoPH–Departments of Public Health; CIMS–Case and incident management system; PH–Public Health.

similar training and ages to myself . . . it'd be just that, "it'll be great . . . we do need an integrated system . . . We need a newer system to do our job more efficiently"' (Participant B), with another participant stating that it was not discussed informally, 'not directly in my area' (Participant A). Senior staff had received updates via management meetings and further knowledge of the project was obtained by some participants through a meeting and demonstration of a candidate system that was replicated in all DoPHs nationally and this appeared to the interviewer to generate enthusiasm for the planned change.

'They spoke to everybody in the department . . . They gave us the same program as they gave to the other departments . . . an overview of the system and it looks fabulous . . . it looks like it has great features' (Participant F).

**Varied perceptions of levels of awareness & knowledge.**   Participants stated awareness of the plan to introduce a CIMS ranged from limited to very aware, with high awareness deriving from local working relationships, membership of working groups and observation of a system demonstration. There was an occasional contrast between the way that some participants perceived their colleagues' awareness and understanding of the planned change, 'strong awareness . . . strong understanding' (Participant G), and others' self-evaluated awareness and understanding—'I knew there was a plan to introduce [a digital system], but I didn't know what it related to' (Participant D).

**Positive but frustrated attitude with negative perceptions of others' attitudes.**   The dominant attitude to the planned change to CIM practice was positive but with some frustration around implementation timing and a perception of negative attitudes of other colleagues. Generally, participants expressed a clear understanding of the need for a digital solution for CIM and disease surveillance. Participants felt excited and eager to get a new system with some shocked that such a system didn't exist before. However, there was frustration it was taking so long and even a loss of hope and scepticism that it would happen soon or at all. The need for clear timelines, or a roadmap, and communication was stressed.

'I think it's been spoken about for so long that there is a little bit of hope being lost that it is going to happen. So that there is a road map or that there is something that might keep . . . the team engaged and ready for this.' (Participant F)

In parallel, they considered the problem needed a good solution and that good implementation takes time but that some people in the DoPH expected the change to 'happen overnight' (Participant A). One participant said they were 'ambivalent', in that any new system 'takes a bit of time to get used to' (Participant E).

More negative perceptions related to other colleagues or the wider network of DoPHs—that people could feel anxious, daunted, and even fearful of the change and that some would be resistant, thinking that they had managed without a national digital system until now or were unsure that the proposed solution would meet the needs of their specific role. It was considered that age, technical ability and comfort with digital systems may determine how challenging the change would be for some.

**Perceived benefits to user & service.**   All participants positively described the benefits a national digital CIMS would bring to individual work, (benefits for the user) and the wider work of the DoPH (benefits to public service) with one caveating 'it's a big positive, if it's done right' (Participant E).

Perceived benefits for the user were an all-in-one system for managing outbreaks that would facilitate more efficient, streamlined workflows that were less frustrating. It would also facilitate easier management of case handovers and create an opportunity for out of hours cover from other DoPH due to the digital and standardised nature of the system. However, it was noted that replacing a manual system with a digital one doesn't necessarily mean less paperwork if digital systems are not integrated.

'there's so many different things used on a patient and..not all of them feed into each other . . . there's still is a lot of paperwork and a lot of handwriting.. the information that it

gathers, it's still manually documented for the large part. They [the systems] don't speak to each other . . . which is a shame' (Participant F)

For the wider work of Public Health protection, a CIMS would support evidence based Public Health practice. This included outbreak management that provided a real-time, geo-tagged, national picture for better focusing of resources, and integration of the CIMS with other health and environmental data reporting systems. Thinking on sustainability, it was suggested that the system would be flexible and would grow with the needs of the community and the users. Other benefits that a CIMS could bring were data use for policy development, modelling health impacts of climate change, and overall cost efficiency that can result from workplace efficiencies but also preventing spread of disease and the associated cost to the health system and people's lives. In relation to public health engagement, a CIMS with built in data visualisation options, would facilitate easier reporting, better public communication, and the building of trust and health literacy in the population. Reflecting more on the future of Public Health as profession, one participant considered that such a standardised system could enable more inclusivity of who can contribute to public health protection.

**Uncertainty about the technological solution.**   While the interview aimed to explore factors outside the design of the system that could be harnessed to make people feel more ready to adopt the technology, 7/8 participants expressed uncertainty about system specifications and what had already been considered. Individuals' concerns related to needs of specific roles and previous experiences of technology development. Many were hopeful but uncertain about the capacity of the technology to integrate with local and national systems (laboratory, hospital and surveillance reporting systems), the inclusion of historical epidemiological data, who else should be included as future users (specialty clinics, reference laboratories) in the implementation plan, the degree of privacy it could ensure and whether it could support remote working. Importantly, it was stressed that a new system should replace existing practices for CIM and disease surveillance and that resources would not stretch to running multiple systems concurrently.

'we don't want to be running two systems. We wouldn't . . . have the ability or the resources . . . to be running dual system[s] . . . that will be the number one . . . concern . . . the new system, is it going to be all or nothing? . . . are we going to be putting all our diseases in there or are we going to be . . . expected to run and operate two systems, that are really collecting very much the same data' (Participant C).

## Current readiness & willingness to adopt a national CIMS

In the category of readiness and willingness to adopt a national CIMS, the following themes were identified: varied departmental readiness and willingness; perception other departments will be willing but less ready to adopt; and barriers and enablers to readiness and willingness (Fig 6).

**Varied departmental readiness & willingness.**   Participants' views on departmental readiness and willingness to adopt a CIMS were wide-ranging: 'everybody's very ready' (Participant F) to 'not particularly ready' (Participant B) and some people will not be willing. Some participants did not feel comfortable speaking for the feelings of their department but gave their perception of others willingness based on their own experiences, noting that some might be less willing because 'I've been in that position myself in the past where what I have works fine, could be better, but it's easier than learning something new' (Participant E). Participants in managerial positions more commonly expressed the readiness of the department as ready or very ready. In contrast, more junior staff expressed that some people will be willing and others not—'I know my immediate colleagues, . . . they're open and willing for the change' and

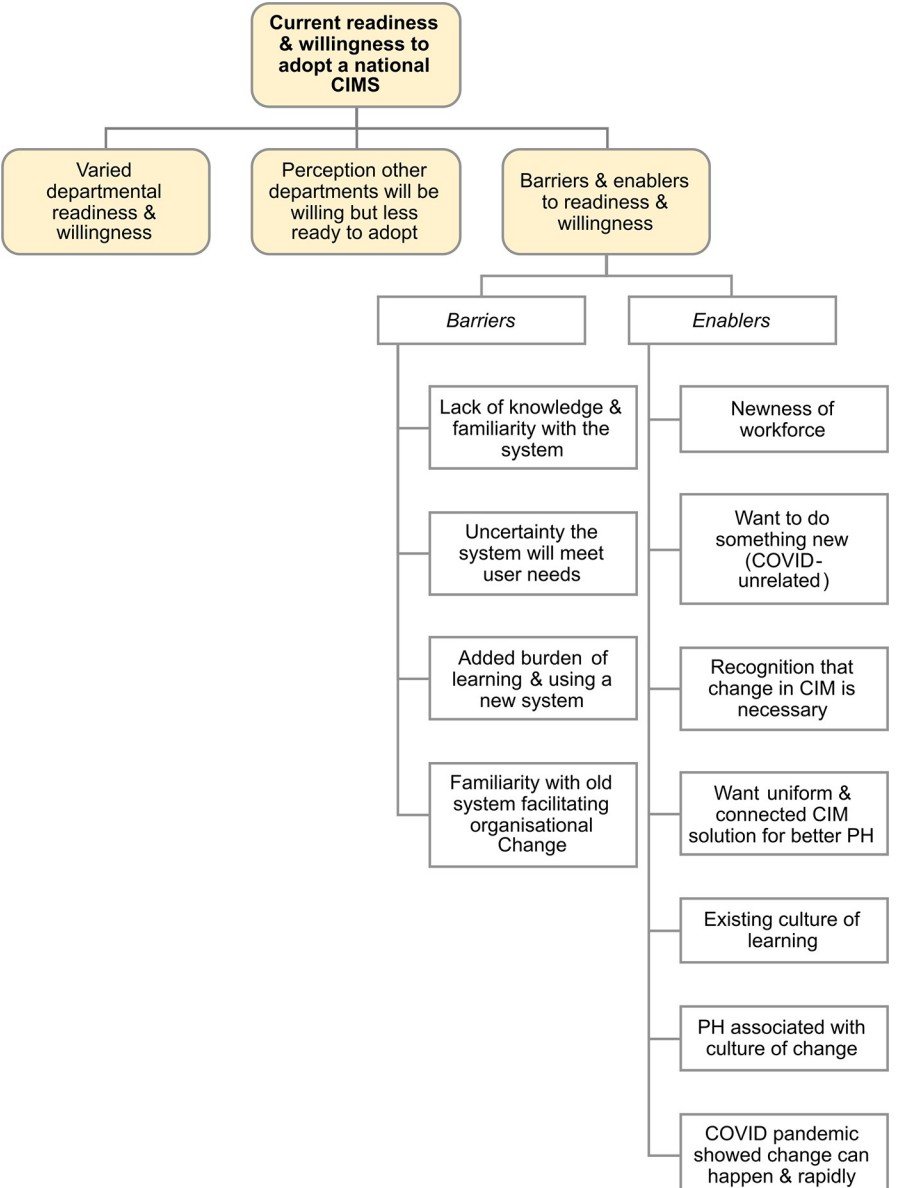

**Fig 6. Summary of the themes and subthemes in current readiness and willingness to adopt a national CIMS.**
Current readiness and willingness to adopt a national CIMS has 3 main themes. Barriers and enablers of readiness and willingness are organised by *barriers* and *enablers*. Illustrative quotes for subthemes of barriers and enablers of readiness and willingness are presented in the supplementary information (S7 Table). PH- Public Health; CIM–case and incident management; CIMS–case and incident management system; COVID–Coronavirus disease.

'I'm sure there's people . . . that'll rail against it'. In contrast, others were 'very willing [but] probably getting frustrated that it's taking so long' (Participant A). Overall, there was a positive perception of the department's potential readiness and willingness for change, and this potential was followed up with the caveats, 'they're very willing if they know the supports are going to be readily there' (Participant F), and 'if people see the reason why . . . and the improvements that [the CIMS] can lead to' (Participant D).

**Perception other departments will be willing but less ready to adopt.** Medical, nursing and administrative participants acknowledged that they had little connection to the wider network of public health departments, but participants considered that other departments would be willing to change as there was already a program of reform in place and a general understanding from the pandemic that better systems were needed.

'the buzzwords are "it's public health reform", and . . . part of this reform includes reforming our systems and the way we work . . . I would think that they would be willing to have a system that's . . . common to us all' (Participant F).

'the recognition that the system that's in place isn't ideal or optimal . . . We realize that . . . there must be better ways to capture the information that we're capturing and so yeah I think it's out of a necessity really that people are ready' (Participant H).

Additionally, it was noted that other departments could be comparatively less ready, with one participant suggesting this may be due to lack of knowledge about digital OMSs in general, 'it might be more overwhelming for the departments who don't have this . . . pre-existing knowledge of how OMSs work' (Participant G) and another, through discussion with colleagues in another department, noted that 'the attention isn't on [OMSs] yet' (Participant A).

**Barriers & enablers to readiness and willingness.** Perceptions of less willingness to adopt the new technology stemmed from the observed barrier of lack of knowledge and familiarity with the system. People were very familiar with the old methods and, while this had facilitated the reform of Public Health practice to multidisciplinary teams–organisational change —with the COVID-19 pandemic the departments had experienced substantial changes in recent times and it was thought that people wanted to hang on to the familiar with some wishing to return to 'life as it was' (Participant D). Other reluctance stemmed from lack of certainty on technology readiness and uncertainty it will meet user needs. Furthermore, learning and using a new system would increase the burden of work, and this would be exacerbated if dual systems (old and new) are in use concurrently.

In contrast, participants emphasised that the newness of the department, the large number of new colleagues who have recently joined, was an enabler of departmental readiness. This new cohort was perceived to have 'an appetite for change' and were 'hungry to learn' (Participant A). Self-identified members of this cohort expressed that now was a good time for change as 'we're not jaded yet' and that 'we're just going with the flow and going with the changes'. One participant noted that 'people don't like the idea of being stagnant', wanting to do something other than deal with the COVID-19 pandemic. Participants considered that the department had a 'learning culture' and people were excited to be 'providing really skilled work using technology'. Willingness was underpinned by the recognition that a change in the practice of CIM was necessary and they wanted a standardised, uniform, and connected (integrated) system for health protection. Interestingly, it was observed by some participants that everyone had experienced rapid change during the COVID pandemic and that the agenda and newness that comes with Public Health reform might make the change occur quicker than would normally be the case in this organisation.

'So you have all this change happening, So the momentum of that could be utilized for the new system, particularly when you're getting all these new recruits into . . . this new way of working . . . in the wider reform' (Participant B).

## Discussion

This study provides a snapshot of the attitudes and behavioural intent, through perceptions of readiness and willingness, of a healthcare workforce who are beginning a digital transformation journey in public health protection. Informed by the perspectives of a diverse set of future users—belonging to administrative, nursing, medical, surveillance, and managerial and staff grade roles—it describes a rich picture of future users' readiness and willingness to adopt new technology for CIM, prior to system procurement and implementation. Critically, it builds our understanding of what underpins future users' attitudes and behavioural intent and why futures users feel ready and willing or not ready or willing to adopt a national digital system for health protection.

### Summary of findings—The rich picture

For the study participants, change has historically happened to them but they have also had positive experiences which provide valuable insights into preferences and expectations during implementation, specifically around communication, engagement, time required and understanding of the contextual events that affect the experience of change. The study established a current view of CIM in the department as a frustrating activity whose efficiency is hindered by limitations in the existing technologies and practices. There was a clear understanding of the need for change and want for a digital solution to aid performance in infection control at a national and global level. While a CIMS was yet to be developed or procured, it was part of national strategic plans for Public Health. However, staff awareness of the plan and its progression varied widely. The predominant attitude towards the project was positive, where any negativity stemmed from frustration with the lack of communication of realistic timelines, anticipation of digital literacy challenges, and indications that some would be resistant to adoption. Critically, there was uncertainty around technology readiness: would the system meet the needs of different users; what were the design specifications; what would be the capacity for integration into existing and future HIS; and how might it integrate with existing practices such as remote working? The above notwithstanding, the majority of users self-evaluated as ready, or very ready, and willing to adopt a new CIMS. However, they were less sure of the readiness and willingness of their colleagues in other departments. They anticipated that, while willing—in that most would recognise the need for a national CIMS as demonstrated during the COVID-19 pandemic—many were not likely to be ready to adopt a new system. Participants speculated that barriers were: lack of familiarity and knowledge of the new system; needing to hold onto familiar practices during organisational and situational change; and feelings of already being under pressure. In contrast, the composition of the department, with many new people and roles, was perceived to be an enabler of readiness and willingness and this was a result of the Public Health reform strategy and rapid expansion during the COVID-19 pandemic.

### Theories of technology adoption

Applying theories of technology adoption, this group showed high 'perceived usefulness' with participants clearly conveying the problems they were having with the current technological solution to CIM, and perceived benefits of a standardised national CIMS. 'Perceived ease of use' refers to how easy future users perceive the new technology is to use and if the benefits exceed the efforts [15]. In the context of this study, a candidate CIMS was trialled in 2017 and another demonstrated in 2022. Participation was inconsistent, however one participant stated that the CIMS appeared 'user friendly'. In addition, when discussing digital literacy and colleagues' readiness for change, it was noted that some would require support, suggesting that

future users anticipate varied perceptions on the ease of use and effort needed to learn the new system. A potential enabler may lie in organisational learning approaches which have been shown to influence perceived ease of use as seen in the introduction of electronic health records into early adopter hospitals [30] and, in the current study, an existing culture of shared learning was highlighted by some participants.

Following the TAM, the 'attitude' of users was positive to the technology but there was frustration about the uncertainty of the timing of the implementation, with many participants noting the need for a roadmap or timeline, albeit a flexible one. However, there still appeared a strong behavioural intent to use the new CIMS as indicated by 'ready' and 'very ready' perceptions of readiness and willingness. This suggests that this cohort of Public Health professionals would likely use the technology when introduced, and willingly, if it meets their needs.

In the UTAUT, social influence directly affects behavioural intent, where social influence refers to 'the degree to which an individual perceives that important others believe he or she should use the new system' [16]. In this study there was limited evidence of social influence beyond the immediate professional peer groups of the participants. There was evidence that some participants had direct relationships to the future implementers, working alongside them, or knowing them from departmental work, but this influence may not be transferable between departments. Notably, within surveillance, medical and nursing staff perspectives, there was evidence of discussions around the need for a better system. This suggests that professional groups may feel some social influence to adopt the technology. In planning the implementation, and applying the theory of DOI [17], persons with social influence, early adopters or opinion leaders who, as a participant suggested, have a positive outlook can influence those struggling with the transition. These early adopters can spread the willingness for adoption of the HIS, as has been seen in other contexts with health professionals [31].

## Barriers & facilitators of technology adoption

A recent systematic review combined with qualitative discussion with experts, identified 77 barriers and 292 success factors for eHealth adoption and implementation [32]. Knowledge/ exposure to eHealth emerged as the dominant barrier after systematic review [32], equating to lack of knowledge and familiarity with CIMS identified in our study. Additionally, the identified added pressure of learning or using a new system also aligned with the systematic review's findings relating to workload. The review's technical barriers were also represented in our study by uncertainties in the system specifications. In parallel, individual enablers included benefits to the service. There was no identification from Schreweis et al's review that a 'new workforce' was an enabler, while being <50 years old was. However, a recent mixed methods study assessing GPs digital readiness to use 5 digital health technologies in their practice, found that 'time in the job' (< 6 years) was predictive of higher digital readiness [33], a finding that was qualitatively observed here. Hammerton et al also identified that GPs as a professional group, were less digitally ready and this linked to lower ability to problem solve (self-efficacy), needing support for IT issues [33]. On a similar theme, in our study, non-clinical staff expressed higher comfort learning new technologies. Hammerton et al also suggested other influences on readiness, with smaller rural departments being less ready for digital transformation than larger urban sites [33]. This should be examined when determining the digital divide and national readiness prior to implementation.

## Opportunities for organisational change

While the majority of participants considered themselves new to the department, perceptions of need and usefulness of the CIMS seem to be driven by experiences during the COVID-19

pandemic. Simultaneously, Public Health reform was both an enabler of readiness, creating a newly employed cohort eager for change, and a barrier to readiness, with individuals' reliance on familiarity with old methods as they experienced dramatic changes to how they worked (i.e. the Public Health reform shift to multidisciplinary teams). In 2002, Staudenmayer and colleagues demonstrated that work rhythm-altering 'temporal shifts' change the collective experience of time and enable organisational change [34]. They proposed that such disruptions 'act as a trigger for change, provide the resources for change, are the coordinating mechanism for change, and are a credible symbol of the need for change' [34]. COVID-19, discrete events in Public Health reform, and emerging diseases such as Mpox, may represent opportunities for such temporal shifts, allowing brief pauses to reflect and adjustments to be made to tackle both emergent and engrained challenges in the practice of public health protection.

## Strengths & limitations of the study

The strengths of this study lie in its timing in the broader implementation plan and its qualitative approach. Key informant interviews with members representing each future user group—administrative, nursing, medical and surveillance, managers and staff grades—facilitated broad and deep understanding of the problems the technology needs to address, future users' concerns about the new technology and their present view of readiness and willingness to adopt a new HIS into public health protection practice. Critically, this was done before procurement and implementation enabling the identification of specific challenges that can be addressed in the both the system design and the implementation process. This should enable project implementers to develop solutions to address the needs and concerns of future users and develop a plan that enhances the organisation's readiness, ensuring that the technology will be successfully and sustainably adopted.

However, the study has some limitations. Primarily, as a small department that grew rapidly during the COVID-19 pandemic, perspectives from public health professionals with experience in public health protection prior to COVID-19 are underrepresented. This group may have a different understanding and awareness of the problem the technology is addressing and have a different attitude to the upcoming change compared to those who joined the department more recently. Thus, the participants' assessments of readiness and willingness are likely not generalisable beyond the 'new' cohort. Additionally, the study was performed in a single department, in which members of the national Outbreak CIMS project are based. Thus, this sample of participants may represent a 'primed' group—they may have a greater knowledge and understanding of the planned CIMS than colleagues in the wider network of DoPHs. The awareness, attitude, and behavioural intent of this department may therefore be different to the wider network of DoPHs.

## Conclusion

This study provides snapshot of the current readiness and willingness of DoPH members to a adopt a national digital CIMS into their work practice. It identifies being 'new' and eager for change as an enabling factor and the 'unknown'–lack of knowledge and familiarity with the system–as the main barrier. Developing an implementation strategy that alleviates these barriers and enhances readiness and willingness will generate confidence among the Public Health protection workforce to move forward with what will be a historic change in the practice of public health protection in Ireland.

Future work will examine future users' perspectives on how to enhance readiness and willingness in the wider network of DoPH through implementation, based on past experiences of change. Together, this work may inform the design of a national survey to elicit a full

assessment of organisational readiness to adopt a CIMS and determination of the wider barriers and enablers to readiness which will inform the implementation plan.

## Supporting information

**S1 Text. Completed Standards for Reporting of Qualitative Research Checklist.**
(DOCX)

**S2 Text. Interview guide for semi-structured key informant interviews.**
(DOCX)

**S1 Table. Perceptions of change from past experiences.**
(PDF)

**S2 Table. Perceived weaknesses of CIM practice.**
(PDF)

**S3 Table. Perceived strengths of CIM practice.**
(PDF)

**S4 Table. Positive but frustrated attitude with negative perceptions of others' attitudes.**
(PDF)

**S5 Table. Perceived user & service benefits.**
(PDF)

**S6 Table. Uncertainty about the technological solution.**
(PDF)

**S7 Table. Barriers & enablers to readiness & willingness to adopt a CIMS.**
(PDF)

## Acknowledgments

The authors would like to acknowledge the support and advice of Dr Niamh Cummins, and Philip Ryan and Stephanie Lee for their help in submitting the ethics application. Special thanks to Dr Angel Mthunzi, Ursula Morley, Dr Conor Mulvagh and Nutsa Burdurladze for their time and feedback on the preliminary interview guide.

## Author Contributions

**Conceptualization:** Dorothea Ogmore Tilley, Brian McKeon, Marie Casey.

**Data curation:** Dorothea Ogmore Tilley.

**Formal analysis:** Dorothea Ogmore Tilley.

**Investigation:** Dorothea Ogmore Tilley.

**Methodology:** Dorothea Ogmore Tilley, Brian McKeon.

**Project administration:** Dorothea Ogmore Tilley, Brian McKeon, Nuha Ibrahim, Marie Casey.

**Supervision:** Brian McKeon, Nuha Ibrahim, Marie Casey.

**Validation:** Brian McKeon.

**Visualization:** Dorothea Ogmore Tilley, Nuha Ibrahim.

**Writing – original draft:** Dorothea Ogmore Tilley.

**Writing – review & editing:** Dorothea Ogmore Tilley, Brian McKeon, Nuha Ibrahim, Stephen H-F Macdonald, Marie Casey.

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
