## [Decision Letter · Decision Letter 0]

10 Nov 2023

PDIG-D-23-00266

A snapshot on a journey from frustration to readiness – a qualitative pre-implementation exploration of readiness for technology adoption in Public Health Protection in Ireland

PLOS Digital Health

Dear Dr. Tilley,

Thank you for submitting your manuscript to PLOS Digital Health. After careful consideration, we feel that it has merit but does not fully meet PLOS Digital Health's publication criteria as it currently stands. Therefore, we invite you to submit a revised version of the manuscript that addresses the points raised during the review process.

Please submit your revised manuscript within 30 days Dec 10 2023 11:59PM. If you will need more time than this to complete your revisions, please reply to this message or contact the journal office at digitalhealth@plos.org. Please include the following items when submitting your revised manuscript:

We look forward to receiving your revised manuscript.

Kind regards,

Yuan Lai, Ph.D.

Academic Editor

PLOS Digital Health

Journal Requirements:

Additional Editor Comments (if provided):

Reviewers' comments:

Reviewer's Responses to Questions

**Comments to the Author**

1. Does this manuscript meet PLOS Digital Health’s publication criteria? Is the manuscript technically sound, and do the data support the conclusions? The manuscript must describe methodologically and ethically rigorous research with conclusions that are appropriately drawn based on the data presented.

Reviewer #1: Yes

Reviewer #2: Partly

2. Has the statistical analysis been performed appropriately and rigorously?

Reviewer #1: Yes

Reviewer #2: Yes

3. Have the authors made all data underlying the findings in their manuscript fully available (please refer to the Data Availability Statement at the start of the manuscript PDF file)?

Reviewer #1: Yes

Reviewer #2: No

4. Is the manuscript presented in an intelligible fashion and written in standard English?

Reviewer #1: Yes

Reviewer #2: Yes

5. Review Comments to the Author

Reviewer #1: The study was done on an important public health issue as Ireland is still in a plight of pervasive technology adoption in public health. The title is really great; it is descriptive and explanatory, and it also fits or goes with the scope of PLOS Digital Health. However, the manuscript has too many caveats to be considered for publication as it stands now.

The full comment is uploaded as a word document.

Reviewer #2: Many thanks for the opportunity to review this manuscript which I have read with interest. I have a few suggestions for ways that the authors might want to improve to aid readability and understanding of this important work.

I suggest to move the table with the number of participant earlier into the methods section to summarise the section which describes participants, page 8.

I suggest the authors set out clearly in the introduction, the aims and objectives of the study and then perhaps reflect at the end of the manuscript within the discussion any reflections against original objectives.

The themes are nicely presented in the thematic map visual on Page 14 and throughout the results section in the for of the sub theme map visuals. However there is inconsistency between the main thematic map on page 14, the sub theme visuals and the section headings and text that follows throughout the results section. This makes the results section difficult to follow, with the visuals not flowing clearly into the sections, as well as theme headings which do not reflect the text that follows. I therefore suggest the results section would benefit from restructuring to ensure it is easy to read and flows logically from the first visual presentation of the themes and subthemes. Ensure headings are consistent throughout and reflect themes and the sub themes as they have been presented. Start each section with the visual (perhaps with an introduction) and then ensure the subheadings and description of findings flow from the thematic visual / flow map. Furthermore, make sure all headings are describing the themes you have presented - e.g. Digital literacy is described well, but I can’t find this presented on any of the flow maps as a theme.

Your page 14 thematic map has arrows reflecting relationships / dependencies between themes, For example, I could imagine that past experiences of change might influence readiness and willingness to implement new technology? I would argue that there are more linkages then presented. I suggest the approach taken to identify these dependencies / linkages were identified or is it the authors opinions? 

Finally, I suggest you break up the discussion section with sub-headings to aid readability.

Many thanks for the opportunity to review and provide feedback to this important work, best of luck.

6. PLOS authors have the option to publish the peer review history of their article (what does this mean?). If published, this will include your full peer review and any attached files.

**Do you want your identity to be public for this peer review?** For information about this choice, including consent withdrawal, please see our Privacy Policy.

Reviewer #1: No

Reviewer #2: Yes: Shoshana Bloom

---

## [Editor Report · Decision Letter 1]

19 Jan 2024

A snapshot on a journey from frustration to readiness – a qualitative pre-implementation exploration of readiness for technology adoption in Public Health Protection in Ireland

PDIG-D-23-00266R1

Dear Dr Tilley,

We are pleased to inform you that your manuscript 'A snapshot on a journey from frustration to readiness – a qualitative pre-implementation exploration of readiness for technology adoption in Public Health Protection in Ireland' has been provisionally accepted for publication in PLOS Digital Health.

Best regards,

Yuan Lai, Ph.D.

Academic Editor

PLOS Digital Health